# An Improved YOLOv2 for Vehicle Detection

**DOI:** 10.3390/s18124272

**Published:** 2018-12-04

**Authors:** Jun Sang, Zhongyuan Wu, Pei Guo, Haibo Hu, Hong Xiang, Qian Zhang, Bin Cai

**Affiliations:** 1Key Laboratory of Dependable Service Computing in Cyber Physical Society of Ministry of Education, Chongqing University, Chongqing 40004, China; zhongyuanw@cqu.edu.cn (Z.W.); pei.guo@cqu.edu.cn (P.G.); hbhu@cqu.edu.cn (H.H.); xianghong@cqu.edu.cn (H.X.); zhngqn@cqu.edu.cn (Q.Z.); caibin@cqu.edu.cn (B.C.); 2School of Big Data & Software Engineering, Chongqing University, Chongqing 401331, China

**Keywords:** vehicle detection, object detection, YOLOv2, convolutional neural network

## Abstract

Vehicle detection is one of the important applications of object detection in intelligent transportation systems. It aims to extract specific vehicle-type information from pictures or videos containing vehicles. To solve the problems of existing vehicle detection, such as the lack of vehicle-type recognition, low detection accuracy, and slow speed, a new vehicle detection model YOLOv2_Vehicle based on YOLOv2 is proposed in this paper. The k-means++ clustering algorithm was used to cluster the vehicle bounding boxes on the training dataset, and six anchor boxes with different sizes were selected. Considering that the different scales of the vehicles may influence the vehicle detection model, normalization was applied to improve the loss calculation method for length and width of bounding boxes. To improve the feature extraction ability of the network, the multi-layer feature fusion strategy was adopted, and the repeated convolution layers in high layers were removed. The experimental results on the Beijing Institute of Technology (BIT)-Vehicle validation dataset demonstrated that the mean Average Precision (mAP) could reach 94.78%. The proposed model also showed excellent generalization ability on the CompCars test dataset, where the “vehicle face” is quite different from the training dataset. With the comparison experiments, it was proven that the proposed method is effective for vehicle detection. In addition, with network visualization, the proposed model showed excellent feature extraction ability.

## 1. Introduction

In order to properly solve urban traffic problems and overcome the existing disadvantages, such as the lack of enough vehicle information and the low accuracy of vehicle information retrieval, intelligent transportation was strongly developed. As an indispensable part of this method, vehicle detection is widely studied by researchers all over the world.

At present, the common vehicle detection methods can be divided into two categories: traditional methods and deep-learning-based methods. The traditional methods refer to traditional machine learning algorithms. References [1,2,3] adopted the histogram of oriented gradient (HOG) method to extract vehicle-type features in images, and then classified those features using the support vector machine (SVM), thus achieving vehicle detection. In Reference [4], a deformable part model (DPM) was proposed for vehicle detection and obtained a good result. Although the accuracy of vehicle positioning and type recognition of those traditional machine-learning-based methods are acceptable, such methods include very complex steps, need high human involvement, and cost too much time. Thus, those methods are not suitable for practical application scenarios. In recent years, deep learning [5] became a very popular research direction. The deep-learning-based object detection and recognition methods usually show better performance than that of the traditional methods [6,7,8]. To obtain richer features of vehicles, References [9,10,11] researched vehicle detection using convolutional neural networks (CNNs). Such methods do not need human-involved feature design, while only a large number of tagged vehicle images are used to train the network with supervision before the network can learn the vehicle-type features automatically. In Reference [12], the network was pre-trained using the unsupervised method with sparse coding, and the vehicle classification was then conducted by softmax. R-CNN [13] was the first model in the field for deep learning object detection. The algorithm uses a selective search to generate a region of interest, which creates a deep learning object detection method based on the region proposal, as implemented in SPP-net [14], Fast R-CNN [15], Faster R-CNN [16], and R-FCN [17]. Reference [18] proposed an adaptive neural network, which extracted features of different scales by dividing the last layer into several networks. It is superior to other traditional methods. Reference [19] improved CNN and proposed a unified multi-scale deep CNN (MS-CNN), which was used to conduct vehicle detection by dividing it into two sub-networks, namely the region proposal network and the detection network. The results showed that the accuracy was improved, and the memory and computation were improved greatly. Furthermore, the MS-CNN can conduct detection with at a rate of frames per second. References [20,21] applied the Faster R-CNN-based method to vehicle detection, and achieved good detection. Reference [22] combined Faster R-CNN, VGG16, and ResNet-152 for vehicle detection, which achieved good vehicle detection accuracy, although the speed was slow and could not satisfy the requirements for real-time vehicle detection. In general, the speed of methods based on deep learning are slow, and cannot meet the real-time requirement. Detection accuracy and generation ability need improvement. Hence, to improve the speed and accuracy of region-based object detection methods, Redmon et al. converted direct object detection to regression, and proposed the end-to-end object detection method YOLO [23]. In 2017, Redmon et al. proposed the YOLOv2 [24] object detection model, which greatly improved the speed of object detection while keeping the detection accuracy.

To improve the vehicle detection accuracy, speed, and generalization ability, a new vehicle detection model based on YOLOv2 is proposed in this paper. The k-means++ [25] clustering algorithm was used to select six anchor boxes with different sizes in the training dataset. To decrease the influence of the vehicles with different sizes on the vehicle detection model, the loss function was improved with normalization. Also, the YOLOv2_Vehicle network was designed by adopting the multi-layer feature fusion strategy and removing the repeated convolutional layer in high layers to improve the feature extraction ability of the network.

## 2. Brief Introduction of YOLO and YOLOv2

In 2016, Redmon et al. proposed the end-to-end object detection method YOLO [23]. As shown in Figure 1, YOLO divides the image into S × S grids and predicts B bounding box and C class probability for each grid cell. Each bounding box consists of five predictions: *w*, *h*, *x*, *y*, and object confidence. The values of *w* and *h* represent the width and height of the box relative to the whole image. The values of (*x*, *y*) represent the center coordinates of the box relative to the bounds of the grid cell. The object confidence represents the reliability of existing object in the box, which is defined as.
(1)$$Confidence=\mathrm{Pr}(object)\times IOU_{pred}^{truth}$$

In Equation (1), Pr(object) represents the probability of the object falling into the current grid cell. $IOU_{pred}^{truth}$ represents the intersection over union (IOU) of the predicted bounding box and the real box.

Then, most bounding boxes with low object confidence under the given threshold are removed. Finally, the non-maximum suppression (NMS) [26] method is applied to eliminate redundant bounding boxes.

To improve the YOLO prediction accuracy, Redmon et al. proposed a new version YOLOv2 in 2017 [24]. A new network structure Darknet-19 was designed by removing the full connection layers of the network, and batch normalization [27] was applied to each layer. Referring to the anchor mechanism of Faster R-CNN, k-means clustering was used to obtain the anchor boxes. In addition, the predicted boxes were retrained with direct prediction. Compared with YOLO, YOLOv2 greatly improves the accuracy and speed of object detection.

However, as a general object detection model, YOLOv2 is applicable to cases where there are a variety of classes to be detected, and the differences among the classes are large, such as persons, horses, and bicycles. However, for vehicle detection, the differences are usually in local areas, such as tires, headlights, and so on. Therefore, to better detect vehicles, this paper proposes an improved YOLOv2 vehicle detection method, and obtained good performance on the validation dataset and another dataset where the “vehicle face” was different from the training dataset.

## 3. Dataset

In this paper, two vehicle datasets collected from road monitoring, the Beijing Institute of Technology (BIT)-Vehicle [28] and CompCars [29], were used. The BIT-Vehicle dataset was provided by the Beijing Institute of Technology and contains 9580 vehicle images. It includes six vehicle types: sedan, sport-utility vehicle (SUV), microbus, truck, bus, and minivan. The number of images for each type is 5922, 1392, 883, 822, 558, and 476, respectively. The CompCars dataset was provided by Stanford University and consists of two sub-datasets. One dataset involves commercial vehicle model pictures collected from the internet, with 1687 vehicle types. The other involves vehicle pictures collected from road surveillance cameras. CompCars only includes two vehicle types: sedan and SUV, with more than 40,000 images. Both datasets include day scenes and night scenes. In addition, the images in both datasets are on sunny days, and there is no presence of noise background, rain, snow, people, other vehicle types, and so on.

The BIT-Vehicle dataset was divided into a training dataset and validation dataset with the ratio of 8:2, where the numbers of images in the training dataset and validation dataset were 7880 and 1970, respectively. For training and validation, the numbers of nighttime images were about 1000 and 250, respectively. To further study the generalization ability and the characteristics of the proposed model, 800 vehicle images were selected randomly from the second sub-dataset of the CompCars dataset to be used for the test dataset and were annotated manually.

Some images in BIT-Vehicle and CompCars datasets are shown in Figure 2 and Figure 3. There are big differences between these two datasets. However, to further study the generalization ability of the proposed model and compare the performance with other models, it was necessary to use the second sub-dataset of CompCars dataset as the test dataset.

## 4. The Improved YOLO_v2 Vehicle Detection Model

### 4.1. Selection of Anchor Boxes

In this paper, k-means++ clustering was applied to conduct clustering analysis on the size of the vehicle bounding boxes in the BIT-Vehicle training dataset. The numbers and the sizes of anchor boxes suitable for vehicle detection were selected. When implementing k-means++, instead of using the traditional Euclidean distance, the distance function of YOLOv2 was applied. As shown in Equation (2), the IOU was adopted as the evaluation metric, which made the error irrelevant to the sizes of anchor boxes.
(2)d(box,centroid)=1−IOU(box,centroid)

As shown in Figure 4, by analyzing the clustering results, the value of k was finally set to be 6, which meant that six anchor boxes of different sizes would be applied for positioning. The right side of Figure 4 shows the six clustering anchor boxes. From the anchor boxes, it can be seen that some clustering anchor boxes were thin and long, while some were square. Those shapes conformed to the actual shapes of the six vehicle types, while the information regarding the distance from the camera was also included. Thus, using clustering analysis on the training dataset with k-means++, the sizes of the anchor boxes suitable for vehicle detection could be obtained, which may improve positioning accuracy.

### 4.2. Improvement of Loss Function

For vehicle detection, since the vehicle picture was obtained from road surveillance cameras, this meant that the vehicle approached the camera during detection. As shown in Figure 5, when the car is far from the camera, it appears smaller in the picture. When it is closer to the camera, it takes up a larger area in the image. Therefore, even if the vehicle type is identical, the size may be different in the picture.

While training YOLOv2, different object sizes had different effects on the whole model, which resulted in larger errors for larger-sized objects than for smaller-sized objects. In order to reduce this influence, the loss calculation for the width and height of the bounding boxes was improved using normalization. The improved loss function is shown in Equation (3).
(3)$$\begin{array}{c} \lambda_{coord}\sum_{i=0}^{S^{2}}\sum_{j=0}^{B}I_{ij}^{obj}[{\left(x_{i}-{\hat{x}}_{i}\right)}^{2}+{\left(y_{i}-{\hat{y}}_{i}\right)}^{2}]\\ +\lambda_{coord}\sum_{i=0}^{S^{2}}\sum_{j=0}^{B}I_{ij}^{obj}[{\left(\frac{w_{i}-{\hat{w}}_{i}}{{\hat{w}}_{i}}\right)}^{2}+{\left(\frac{h_{i}-{\hat{h}}_{i}}{{\hat{h}}_{i}}\right)}^{2}]\\ +\sum_{i=0}^{S^{2}}\sum_{j=0}^{B}I_{ij}^{obj}{\left(C_{i}-{\hat{C}}_{i}\right)}^{2}\\ +\lambda_{noobj}\sum_{i=0}^{S^{2}}\sum_{j=0}^{B}I_{ij}^{noobj}{\left(C_{i}-{\hat{C}}_{i}\right)}^{2}\\ +\sum_{i=0}^{S^{2}}I_{i}^{obj}\sum_{c\in classes}{\left(p_{i}(c)-{\hat{p}}_{i}(c)\right)}^{2} \end{array}$$
where $x_{i}$ and $y_{i}$ are the center coordinates of the box of the i-th grid cell, $w_{i}$ and $h_{i}$ are the width and height of the box of the i-th grid cell, $C_{i}$ is the confidence of the box of the i-th grid cell, and $p_{i}(c)$ is the class probability of the box of the i-th grid cell. Furthermore, ${\hat{x}}_{i}$, ${\hat{y}}_{i}$, ${\hat{w}}_{i}$, ${\hat{h}}_{i}$, ${\hat{C}}_{i}$, and ${\hat{p}}_{i}(c)$ are the corresponding predictions of $x_{i}$, $y_{i}$, $w_{i}$, $h_{i}$, $C_{i}$, and $p_{i}(c)$; $\lambda_{coord}$ denotes the weight of the coordinate loss, and $\lambda_{noobj}$ denotes the weight of the bounding boxes without objects loss. Finally, $S^{2}$ denotes the S × S grid cells, B denotes the boxes, $I_{i}^{obj}$ denotes whether the object is located in cell i or not, and $I_{ij}^{obj}$ denotes that the j-th box predictor in cell i is “responsible” for that prediction. In Equation (3), the first line calculates the coordinate loss, the second line calculates the bounding box size loss, the third line calculates the bounding box confidence loss with objects, the fourth line calculates the bounding box confidence loss without objects, and the last line calculates the class loss.

As shown in Equation (3), compared with YOLOv2, we used $\frac{w_{i}-{\hat{w}}_{i}}{{\hat{w}}_{i}}$ and $\frac{h_{i}-{\hat{h}}_{i}}{{\hat{h}}_{i}}$ instead of $w_{i}-{\hat{w}}_{i}$ and $h_{i}-{\hat{h}}_{i}$, which may reduce the effect of the difference sizes of the same vehicle type in the picture, potentially optimizing the detection bounding boxes to a certain degree.

### 4.3. Design of Network

**(1) Multi-Layer Feature Fusion.** For vehicle detection, the differences among vehicles usually involve contour, color, lamp shape, tire shape, etc., while, in the CNN, the local features exist in low layers. To make full use of the local information, a multi-layer feature fusion strategy was adopted. As shown in Figure 6, part (a) goes through 3 × 3 and 1 × 1 convolution layers, and is followed by Reorg/4 for down-sampling. Part (b) conducts the same operations, but the down-sampling factor is 2. The purpose of Reorg is to keep the feature maps of those layers the same. Then, the local features of part (a), part (b), and the global features of one layer are fused, which enhances the network understanding of local information, and enables the model to distinguish the tiny differences among vehicle types.

**(2) Removing the Repeated Convolution Layers in High Layers.** A network model such as YOLOv2 is usually designed as a general object detection model. Thus, the number of classes detected by such a network may be high, and the difference among the classes may be large, such as people, apples, cars, houses, etc. For the YOLOv2 network, there are three continuous and repeated 3 × 3 × 1024 convolution layers in high layers. Usually, the repeated convolution operation in high layers can deal with many classes with large differences, such as people and apples. For vehicle detection, the number of vehicle types detected was only six, and the feature differences among the vehicle types were very small. It means that many repeated convolutional layers in high layers may not improve the performance, serving only to make the model more complex. Therefore, we removed the repeated convolutional layers in high layers. As shown in Figure 6, the number of continuous 3 × 3 × 1024 convolution layers was reduced to one. The last layer is marked with black box.

By applying multi-layer feature fusion and removing the repeated convolutional layers in high layers, the YOLOv2_Vehicle network was finally designed. Also, to verify the effectiveness of removing the repeated convolutional layers in high layers, we designed another network Model_Comp for comparison. Compared with YOLOv2, Model_Comp only removed one 3 × 3 × 1024 convolution layer. The specific network structures of YOLOv2, Model_Comp, and YOLOv2_Vehicle are shown in Table 1.

## 5. Experiments

### 5.1. Environment

The hardware environment of the experiment is shown in Table 2. We conducted the experiments on a graphics processing unit (GPU) server. The GPU used was Nvidia Tesla K80, the video memory was 24 GB, and the operating system was Ubuntu 14 with a memory of 64 GB. The models were implemented on the Darknet platform framework.

### 5.2. Results and Analysis

In the experiment, the initial learning rate was 0.001, which was divided by 10 when the epoch reached 60 and 90. The max epoch was set to 160, the batch size was set to 8, and the momentum was set to 0.9. Every 10 epochs, a new input image size was randomly selected for network training. Considering that the down-sampling factor was 32, all randomly selected input image sizes were multiples of 32, where the minimum size was 352 × 352 and the maximum size was 608 × 608. Such a training method enables the final model to better predict the images with different sizes, while the same model can be used for vehicle detection with different resolutions, which may enhance the robustness of the model.

#### 5.2.1. Analysis of Training Stage

Figure 7 shows the average loss curves of the three models during training. The vertical coordinate denotes the average loss, while the horizontal coordinate denotes the quotient between the number of training iterations and the number of GPUs being used for training. From Figure 7, it can be seen that the average loss had a downward trend, and finally tended to be stable at small values. For the three models, the average loss of the YOLOv2_Vehicle model decreased fastest at the beginning, followed by Model_Comp. The main reason was that both Model_Comp and YOLOv2_Vehicle adopted the feature fusion strategy; thus, more local feature information could be obtained, which accelerated the convergence of training. Although the average loss of the YOLOv2_Vehicle model fluctuated during training, it reached the minimum first among the three models, and was the lowest overall. The average loss of Model_Comp also fluctuated, but was lower than that of YOLOv2. Hence, the network of the YOLOv2_Vehicle model could accelerate the convergence of the vehicle dataset, and fit the vehicle detection task better.

During training, the trend of the average IOU of each model also needed to be considered, because it represented the accuracy of the detected bounding boxes. As shown in Figure 8, the average IOU of the three models showed a gradual upward trend. They were all stable between 0.7 and 1, which shows that the three models had a good performance when locating. Although the IOU results of the three models were close, the initial upward trends of Model_Comp and YOLOv2_Vehicle were faster than that of YOLOv2. In particular, the average IOU of YOLOv2_Vehicle quickly reached between 0.6 and 0.7 in the initial stage, while YOLOv2 needed more training time, which also proves that the network of YOLOv2_Vehicle could accelerate the convergence of the vehicle dataset.

From the above analysis of the training stage, it can be concluded that the trends of both average loss and average IOU of the YOLOv2_Vehicle model were better than those of Model_Comp and YOLOv2.

#### 5.2.2. Analysis of Test Stage

The performances of YOLOv2, Model_Comp, and YOLOv2_Vehicle using the BIT-Vehicle validation dataset with a threshold 0.5 were compared using the recall, precision, and average IOU as the evaluation metrics. As can be seen from Table 3, the two models proposed in this paper showed good performance. The recall, precision, and average IOU of both models were superior to those of YOLOv2. Based on the three evaluation metrics, the YOLOv2_Vehicle model was superior.

For object detection, a very important metric for measuring the performance of a model is mAP. As shown in Table 4, the mAP of the YOLOv2_Vehicle model was the highest, reaching 94.78%. The average detection speed was 0.038s, which means that the model could deal with about 26 pictures in 1 s with regards to results in real time. This is important in some monitoring systems, such as intelligent transportation systems. Compared with the model of Reference [22], the results of YOLOv2_Vehicle and Model_Comp were better. All the classes of AP, mAP, and speed of the models based on YOLOv2 were better than those of Reference [22]. In addition, since the model used in Reference [22] was Faster R-CNN and was based on region proposal, the average detection speed was only 0.68s, which is much different from YOLOv2_Vehicle and Model_Comp. Although the classes of AP for YOLOv2, Model_Comp, and YOLOv2_Vehicle were close, most classes of AP for the YOLOv2_Vehicle model were superior. The network of Model_Comp removed a repeated convolution layer in high layers of YOLOv2, which did not affect the Model_Comp performance on vehicle detection, and the result was better than that of YOLOv2. Thus, it confirmed the basis of this paper in the network design stage, i.e., the repeated convolutional layers in high layers are not suitable for situations with a few classes with minute differences. In other words, the operation of removing repeated convolutional layers in high layers was effective for vehicle detection.

Figure 9 shows the detection results of the YOLOv2_Vehicle model. It can be seen that the YOLOv2_Vehicle model had good performance for both single and multiple vehicle detection. Whether it was daytime or night, the abilities of vehicle positioning and type recognition of YOLOv2_Vehicle were not affected, which proves that YOLOv2_Vehicle has strong weather adaptability. In addition, in the three pictures of the third column in Figure 9, there were some incomplete vehicles. However, from the actual detection results, such a situation did not affect the vehicle detection accuracy of the YOLOv2_Vehicle model. It reflects that YOLOv2_Vehicle has the ability to complete vehicle positioning and type recognition with the vehicle’s local information, and reflects the effectiveness of the multi-layer feature fusion strategy.

Both Model_Comp and YOLOv2_Vehicle adopted the feature fusion strategy. To verify the effectiveness of this strategy for vehicle detection and to further compare the performance between Model_Comp and YOLOv2_Vehicle, the CompCars test dataset was used to test and analyze these two models. In total, 800 vehicle images were randomly selected from the second sub-dataset of the CompCars dataset as the test dataset, named Random_Comp. The mAPs of Sedan and SUV were taken as the standard to measure the performance of the model.

As shown in Table 5, the mAP of the YOLOv2_Vehicle model using the Random_Comp dataset was much higher than that of Model_Comp. However, the results of the two models on the Random_Comp dataset were not very good, where the maximum mAP was only 68.19%. The main reason may be that, compared with the BIT-Vehicle dataset used for training, there were almost no similar “vehicle face” images in the Random_Comp dataset, which means that there were large differences between the training dataset and the Random_Comp dataset. However, the purpose of testing with another dataset with a large difference was not to show that the model can definitely achieve ideal results; instead, it was to compare and analyze the performances across models based on the result, so as to further understand the model characteristics. According to Section 4.3, the YOLOv2_Vehicle adopts the method of multi-layer feature fusion, while Model_Comp only adopts single-layer feature fusion. Although the accuracy of YOLOv2_Vehicle and Model_Comp were a little different using the BIT-Vehicle dataset, the YOLOv2_Vehicle model outperformed Model_Comp using the Random_Comp dataset. Also, the numbers of network parameters of YOLOv2_vehicle and Model_Comp were about 3.94 million and 4.14 million, respectively. Obviously, the complexity of YOLOv2_Vehicle was less than that of Model_Comp, demonstrating that YOLOv2_Vehicle has the stronger ability to understand local information of vehicles, has better generalization ability, and is more suitable for vehicle detection. It also verified the basis of this paper, whereby it is effective to adopt the multi-layer feature fusion strategy for vehicle detection. Figure 10 shows the detection results of YOLOv2_Vehicle. The images in the first column are day scenes. YOLOv2_Vehicle can detect vehicles accurately. The images in the last 2 columns are night scenes. YOLOv2_Vehicle also can detect vehicles accurately. YOLOv2_Vehicle has a good performance on vehicle detection.

#### 5.2.3. Visualizing the Network

Usually, the evaluation metrics for vehicle detection are mAP and speed; however, there exists another way to evaluate the model, i.e., by visualizing the network [30]. This method can observe the quality of the features and the ability of the network for extracting features more directly. Taking a road vehicle image (Figure 11) in the BIT-Vehicle dataset, for instance, the visual features of the YOLOv2_Vehicle model were presented and analyzed. Figure 12 shows the first nine feature maps of Figure 11 after passing through the first convolution layer. It can be seen that most feature maps contain vehicle edge information, which indicates that the convolution kernel in the first layer successfully extracted the edge information of the vehicle.

On the other hand, in the deeper layers, the output feature maps were more abstract and fuzzy and the size gradually decreased, which resulted from the multiple convolutions and down-sampling operations. As shown in Figure 13, when the original vehicle image passed through the fifth convolutional layer, the output feature maps became fuzzier and the textures became more complex; however, there were still some local features.

From the comparison between Figure 12 and Figure 13, it can be concluded that the YOLOv2_Vehicle model can extract vehicle features well. In addition, two feature maps advanced gradually and there was no abrupt recession. Thus, the YOLOv2_Vehicle model has good feature extraction ability, and can appropriately pass the good features extracted from early layers to later layers.

## 6. Conclusions

In this paper, by improving YOLOv2, a model called YOLOv2_Vehicle was proposed for vehicle detection. To obtain better anchor boxes, the vehicle bounding boxes on the training dataset were clustered with k-means++ clustering, and six anchor boxes with different sizes were selected. Next, the loss function was improved with normalization to decrease the influence of the different scales of the vehicles. Then, to obtain better feature extraction ability, the YOLOv2_Vehicle network was designed with the multi-layer feature fusion strategy and removal of the repeated convolution layers in high layers. Based on the experimental results, the mAP of YOLOv2_Vehicle could reach 94.78%. Also, the model showed a good generalization ability using a dataset different from the training dataset. Therefore, the proposed network is effective for vehicle detection. The feature extraction ability of VOLOv2_Vehicle was illustrated with network visualization.

Although the model proposed in this paper achieved ideal experimental results, the number of vehicle types and the amount of data are relatively low. In future work, we will collect more actual vehicle data to further study how to improve the accuracy and speed of vehicle detection.

## Figures and Tables

**Figure 1 sensors-18-04272-f001:**
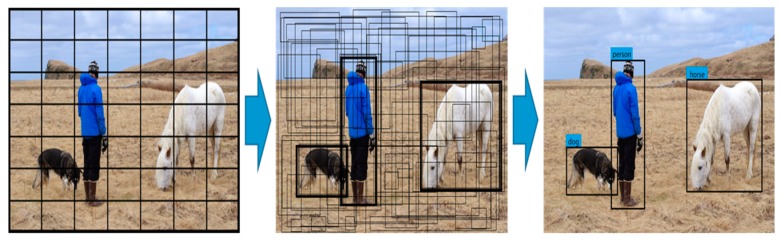
Flowchart of YOLO object detection.

**Figure 2 sensors-18-04272-f002:**
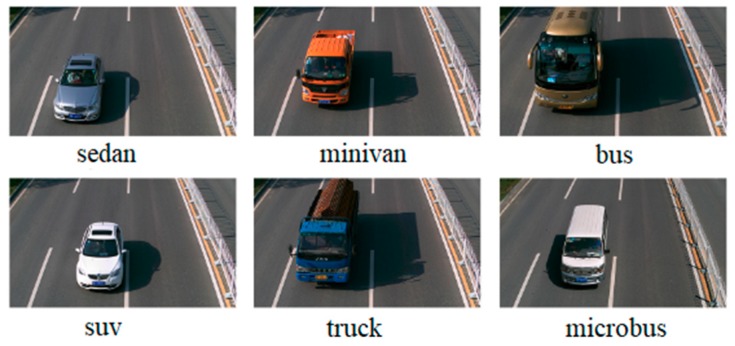
Beijing Institute of Technology (BIT)-Vehicle dataset.

**Figure 3 sensors-18-04272-f003:**
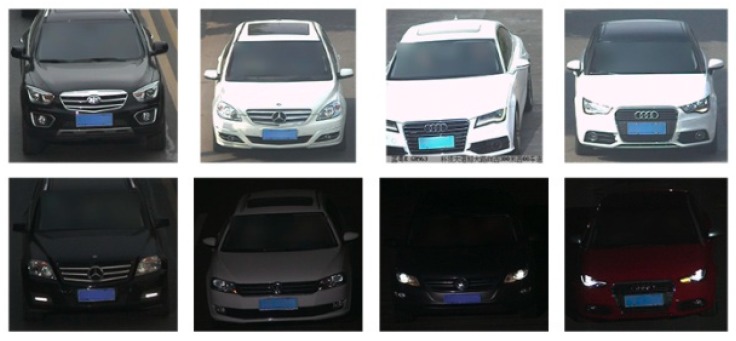
Some images in CompCars dataset.

**Figure 4 sensors-18-04272-f004:**
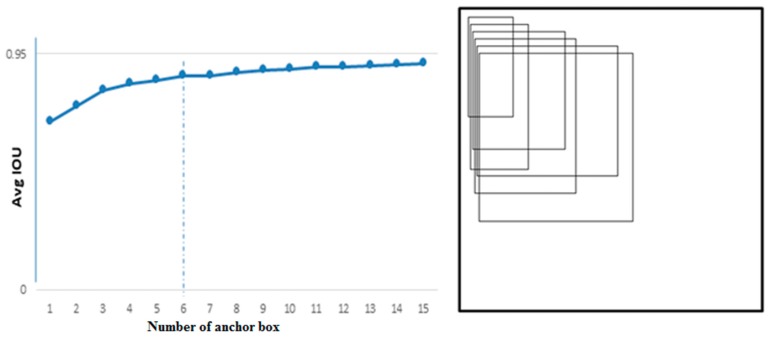
The clustered anchor box information.

**Figure 5 sensors-18-04272-f005:**
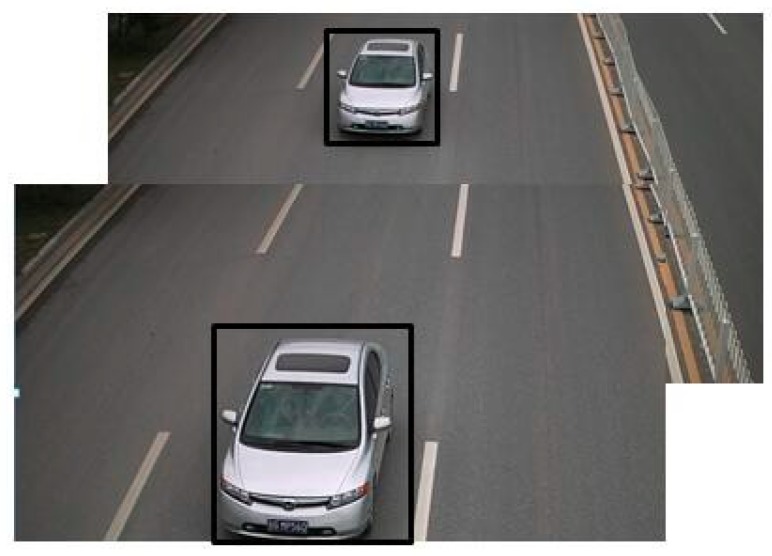
Comparison of the same vehicle with different distance.

**Figure 6 sensors-18-04272-f006:**
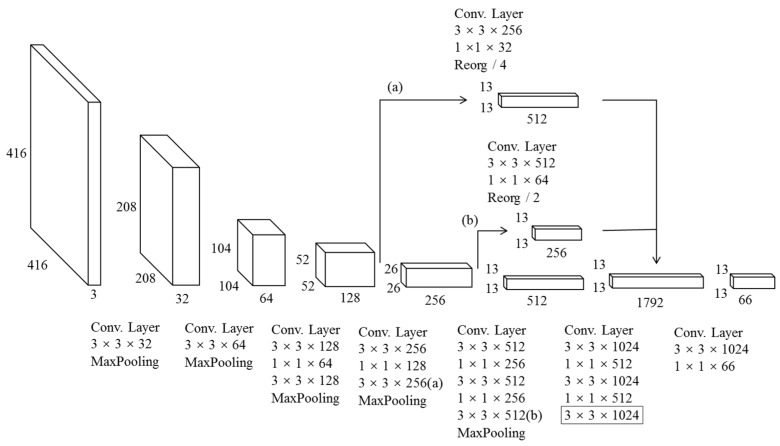
The network structure of the YOLOv2_Vehicle model.

**Figure 7 sensors-18-04272-f007:**
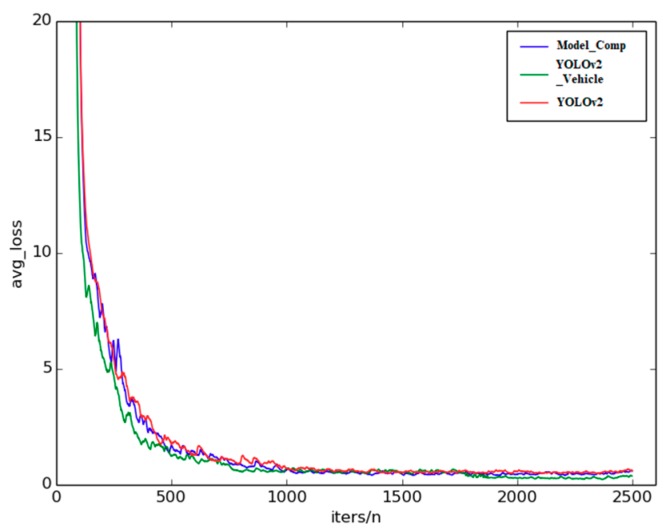
Comparison of the average loss values of the three models.

**Figure 8 sensors-18-04272-f008:**
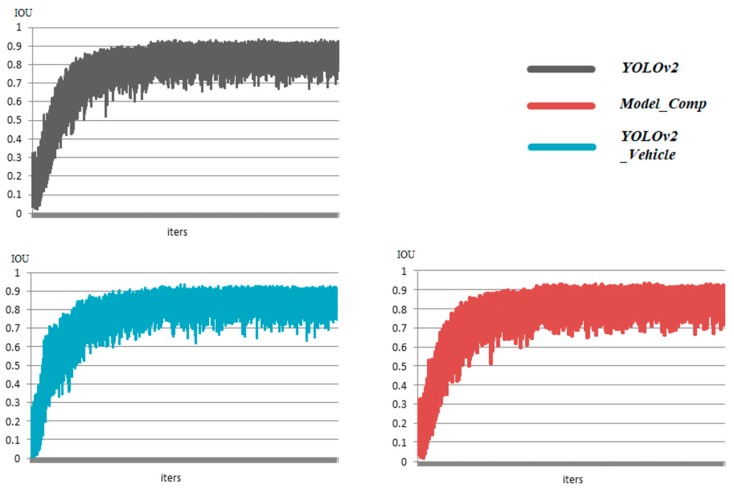
The average intersection over union (IOU) comparison of the three models.

**Figure 9 sensors-18-04272-f009:**
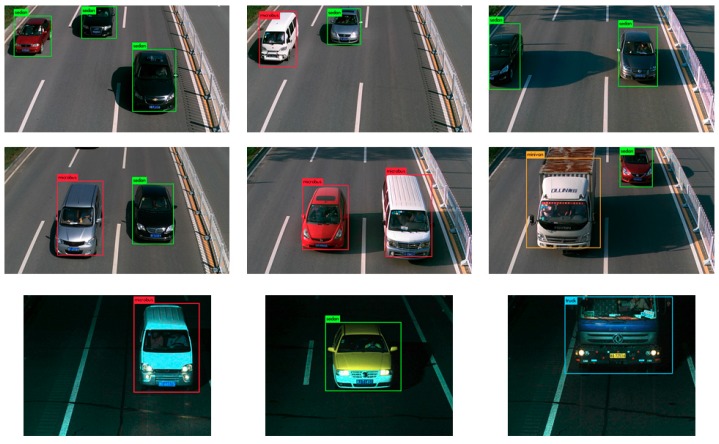
Detection results of YOLOv2_Vehicle using the BIT-Vehicle dataset.

**Figure 10 sensors-18-04272-f010:**
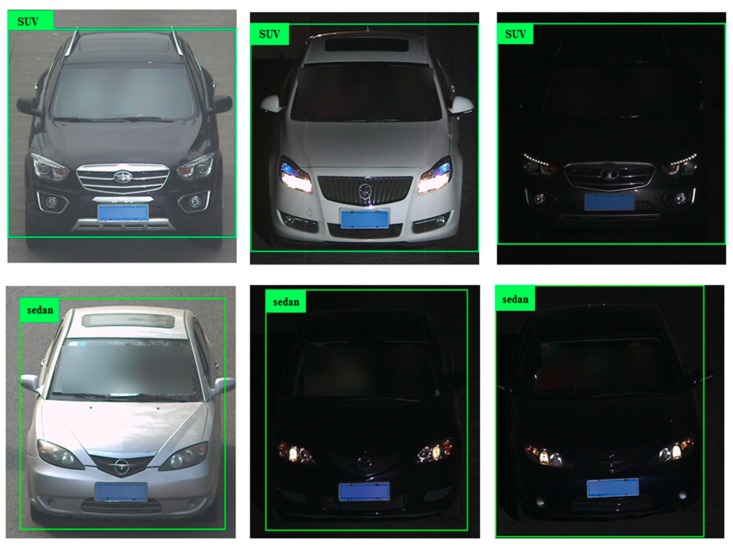
Detection results of the YOLOv2_Vehicle model using the Random_Comp dataset.

**Figure 11 sensors-18-04272-f011:**
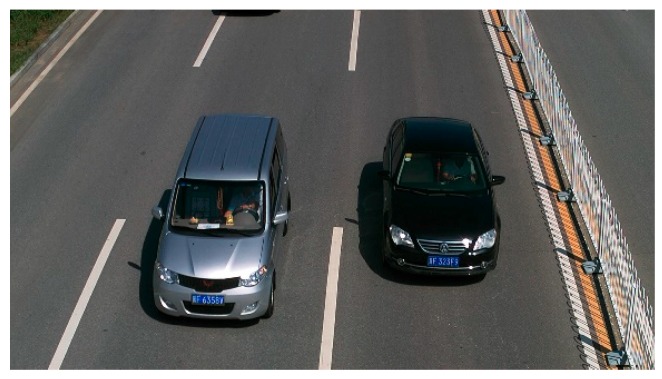
The vehicle image from the BIT-Vehicle dataset.

**Figure 12 sensors-18-04272-f012:**
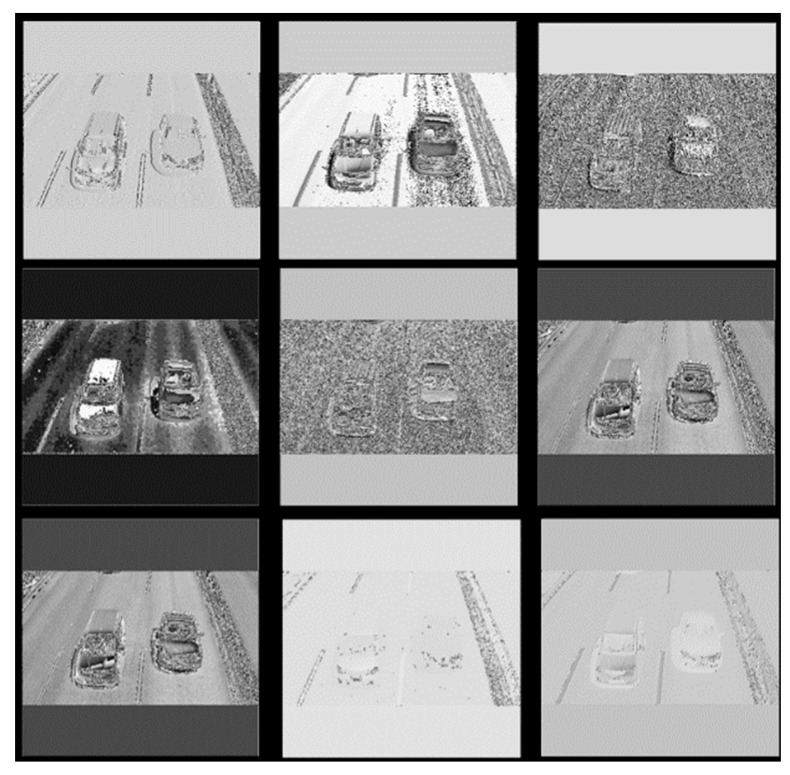
Part feature maps that passed through the first convolution layer in the YOLOv2_vehicle model.

**Figure 13 sensors-18-04272-f013:**
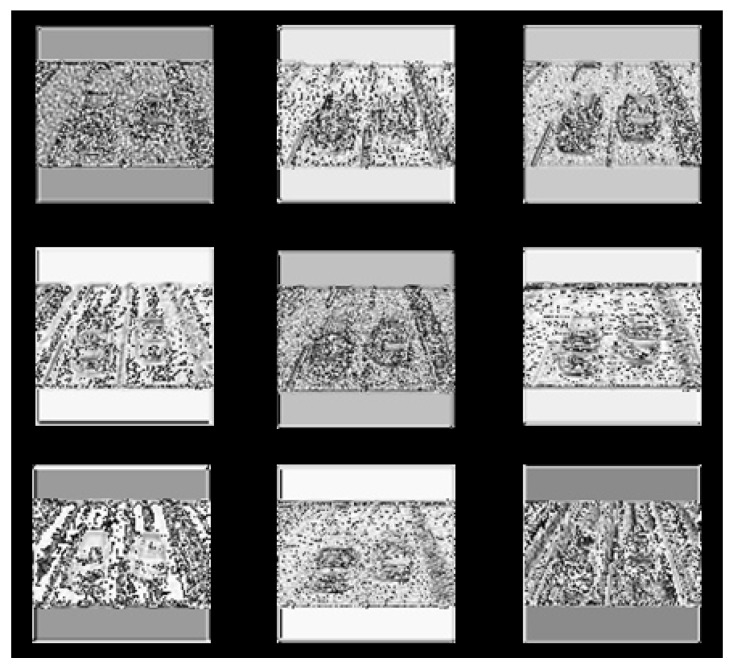
Part feature maps that passed through the fifth convolution layer in the YOLOv2_vehicle model.

**Table 1 sensors-18-04272-t001:** The network structures of YOLOv2, Model_Comp, and YOLOv2_Vehicle.

Layer\Model	YOLOv2	Model_Comp	YOLOv2_Vehicle
0	Conv3-32	Conv3-32	Conv3-32
1	Maxpool/2	Maxpool/2	Maxpool/2
2	Conv3-64	Conv3-64	Conv3-64
3	Maxpool/2	Maxpool/2	Maxpool/2
4	Conv3-128	Conv3-128	Conv3-128
5	Conv1-64	Conv1-64	Conv1-64
6	Conv3-128	Conv3-128	Conv3-128
7	Maxpool/2	Maxpool/2	Maxpool/2
8	Conv3-256	Conv3-256	Conv3-256
9	Conv1-128	Conv1-128	Conv1-128
10	Conv3-256	Conv3-256	Conv3-256
11	Maxpool/2	Maxpool/2	Maxpool/2
12	Conv3-512	Conv3-512	Conv3-512
13	Conv1-256	Conv1-256	Conv1-256
14	Conv3-512	Conv3-512	Conv3-512
15	Conv1-256	Conv1-256	Conv1-256
16	Conv3-512	Conv3-512	Conv3-512
17	Maxpool/2	Maxpool/2	Maxpool/2
18	Conv3-1024	Conv3-1024	Conv3-1024
19	Conv1-512	Conv1-512	Conv1-512
20	Conv3-1024	Conv3-1024	Conv3-1024
21	Conv1-512	Conv1-512	Conv1-512
22	Conv3-1024	Conv3-1024	Conv3-1024
23	Conv3-1024	Conv3-1024	Route 10
24	Conv3-1024	Route 16	Conv3-256
25	Route 16	Conv3-512	Conv3-32
26	Conv1-64	Conv1-64	Reorg/4
27	Reorg/2	Reorg/2	Route 16
28	Route 27 24	Route 27 23	Conv3-512
29	Conv3-1024	Conv3-1024	Conv1-64
30	Conv1-66	Conv1-66	Reorg/2
31	Detection	Detection	Route 30 26 22
32			Conv3-1024
33			Conv1-66
34			Detection

**Table 2 sensors-18-04272-t002:** The hardware environment. GPU—graphics processing unit; CPU—central processing unit.

Hardware	Environment
Computer	GPU server
CPU	Intel(R) Xeon(R) CPU E5-2683 v3 @ 2.00 GHz
GPU	Nvidia Tesla K80 × 4
Memory Size	64 GB

**Table 3 sensors-18-04272-t003:** The recall, precision, and average intersection over union (IOU) when using the Beijing Institute of Technology (BIT)-Vehicle validation dataset.

Model	Recall	Precision	Avg IOU
YOLOv2	99.32%	99.20%	84.43%
Model_Comp	**100%**	99.41%	84.80%
YOLOv2_Vehicle	**100%**	**99.51%**	**89.97%**

**Table 4 sensors-18-04272-t004:** The results using the BIT-Vehicle validation dataset. SUV—sport-utility vehicle.

Model	Bus	Microbus	Minivan	Sedan	SUV	Truck	mAP	s/Img
Model_Comp	97.43%	**94.47%**	90.86%	97.46%	93.05%	91.69%	94.16%	**0.038**
YOLOv2_Vehicle	**97.54%**	93.76%	**92.18%**	98.48%	**94.62%**	**92.09%**	**94.78%**	**0.038**
YOLOv2	96.39%	92.24%	90.61%	**98.57%**	91.49%	90.57%	93.31%	0.045
Faster R-CNN + ResNet [22]	90.62%	94.42%	90.67%	90.63%	91.25%	90.07%	91.28%	0.68

**Table 5 sensors-18-04272-t005:** The mAP using the Random_Comp dataset.

Model	mAP
Model_Comp	54.37%
YOLOv2_Vehicle	68.19%

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
