# Peer review of "An Improved YOLOv2 for Vehicle Detection"

_sensors, 2018, doi:10.3390/s18124272_

Round 1

Reviewer 1 Report

This paper presents a vehicle detection method based on the YOLOv2 network, this paper needs to be improved as the following comments:

1. In section 4.2, the loss calculation of the bounding box was improved by using normalization and the improved loss function is shown in equation (3). Can you clearly define and explain the different terms of the equation? Can the authors more clearly explain about why the loss function of the bounding box can be improved by normalization?

2. Section 4.3 mentioned the reasons on why multi-layer feature fusion was adopted, but the descriptions of the corresponding implementation details were not shown in this paper. Can you provide more information about this issue?

3. In Section 5.2.2, what are the sizes of daytime and nighttime training sets to provide all-day detection results? Can you provide more details about your daytime and nighttime training sets?

4. In the third paragraph of section 5.2.2 - Analysis on test stage, the figure you are talking about should be Figure 9, not Figure 8?

5. In the fifth paragraph of section 5.2.2, line 246 to 247 said that “the number of network parameters of YOLOv2_Vehicle is less than that of Model_Copm”, can you give more clear experiment results to demonstrate this claim?

Minor comments:

1. The font size in Figure 6 is quite small, can you provide a clearer version?

2. There are two section 5.2.1 in section 5.2, and section 5.2.2 is missing, please check.

Reviewer 2 Report

The paper presents a method to uses deep learning for accomplish a vehicle detection. On the one hand, there are lot of papers that provides similar approach, by this reason I considered that the novelty is not so high. On the other hand, the authors present the word performed to extrapolate a CNN from classifying general objects to differentiate more similar ones.

Some paragraphs are repeated in the paper: The in the abstract, some phrases are the same as the ones in last paragraph of the introduction section. Furthermore, the same sentences are repeated in the conclusions. There are to differentiate between abstract and conclusions of the work.

Introduction section is too short to present an adequate state of the art, it must be improved.

To sum up, the work can be considered for publication after some minor revisions

Reviewer 3 Report

This paper presented an variant of YOLOv2 model for vehicle detection. A few changes have been made, such as k-mean and reduced hidden layer in the network. The results demonstrated good improvement against the original YOLOv2 and another compare model. Generally, I think the manuscript has acceptable quality and contributions. Some minor suggestions are to enhance the literature survey and demonstrate its effectiveness and efficiency in real-time vehicle detection. The multi-layer network has large number of parameters to be trained, thus a large number of data is required for such training. 

Reviewer 4 Report

The paper proposes using a modified version of YOLOv2 to detect the vehicle. The authors used k-means++ clustering in order to select anchor boxes. They applied the method to a well-known dataset. In my opinion, the results obtained do not represent a real advance in the field. I do not recommend accepting the paper based on the following.

1-      At line 84 the authors said that “However, as a general object detection model, YOLOv2 does not fit well for vehicle detection.” Why? The authors should be added a deeper explanation about the problems found to detect the vehicle at this point. The introduction needs improvements in order to clarify the existing problems and how the proposed work could overcome them.

2-      There are no explanations about Eq. 3. There is no mention what does each variable mean? Is that Eq. correct?

3-      In the used datasets, are those next features considered? Noise background, rain, snow, people, another car? How would be the behaviour of your system for those features? Have you tested it? Although the dataset is already published, it needs to be pointed out.

4-      I think the authors are overselling the paper too much, as follows:

a)      At line 198: “From above analysis on the training stage, it can be concluded that both of the trends of average loss and average IOU of YOLOv2_Vehicle are better than those of Model_Comp and YOLOv2. This conclusion is very tough to be made based on the presented figures;

b)      At line 209. “Compared with the model of  Ref. 15, the results of YOLOv2_Vehicle and Model_Comp are significantly better, and mAP was 210 improved by 3.5%.” 3.5% is not “significantly better”  results;

c)      There are much more…

5-      The results presented in Table IV are very close as well.  It seems that the found advances are weak.

6-      How did you choose the ideal number of layers? Why not 10 or 20, for instance?

7-      Based on the experimental procedure the authors used a well-known dataset and they mention they selected part for training and another for tests. How were the percentages for each one? It should be clear.

8-      In table 5, what is the comparison was made only by Model_Comp and YOLOV2_Vehicle? What about the others methods which were above presented?

9-      The authors said: “Figure 12 shows the first 9 feature maps of Figure 11 after passing through the first convolution layer.” Posteriorly, the said: “As shown in Figure 13, when the original vehicle image passes through the fifth  convolutional layer, the output feature maps become fuzzier and the textures become more complex, but there are still some obvious local features.” The authors are comparing results from different layers, that in my point of view it constitutes an unfair comparison. If so, it is expected different results.

10-   At line 267, the authors said: “People may not understand those complex features, 267 but the network can.” It doesn’t make any sense once the computer will detect it instead of people. This phrase should be rebuilt.

11-  In the Introduction and Conclusion, the authors said that their system hit 94.78% accuracy that represents a good result. It could be true if the other methods did not hit similar results, for example, 94.42% for R-CNN Microbus.  Much more similar results are found. For this constitutes the biggest problem of the paper. Weak results.

12-   I just pointed out examples that the paper presents weak results and need improvements in the methodology, introduction and experiments. There are more.

Round 2

Reviewer 1 Report

The authors have revised the manuscript according to my comments, and now the current version of the manuscript can be accepted for publication.

Reviewer 4 Report

The authors have significantly improved the manuscript. I have just the following comment before accepting the paper.

1- At lines 244 and 246 the authors use "Ref. 22".  I think they should use only [22] instead.